# Ongoing Bidirectional Feedback between Planning and Assessment in Educational Contexts: A Narrative Review

**DOI:** 10.3390/ijerph191912068

**Published:** 2022-09-23

**Authors:** Manuel Loureiro, Fábio Yuzo Nakamura, Ana Ramos, Patrícia Coutinho, João Ribeiro, Filipe Manuel Clemente, Isabel Mesquita, José Afonso

**Affiliations:** 1Centre for Research, Education, Innovation, and Intervention in Sport (CIFI_2_D), Faculty of Sport of the University of Porto (FADEUP), Rua Dr. Plácido da Costa 91, 4200-450 Porto, Portugal; 2Research Centre in Sports Sciences, Health Sciences and Human Development (CIDESD), University of Maia (ISMAI), Av. Carlos de Oliveira Campos, 4475-690 Maia, Portugal; 3Football Department, Lusophone University of Porto, 4000-098 Porto, Portugal; 4Escola Superior Desporto e Lazer, Instituto Politécnico de Viana do Castelo, Rua Escola Industrial e Comercial de Nun’Álvares, 4900-347 Viana do Castelo, Portugal; 5Research Center in Sports Performance, Recreation, Innovation and Technology (SPRINT), 4960-320 Melgaço, Portugal; 6Instituto de Telecomunicações, Delegação da Covilhã, 1049-001 Lisboa, Portugal

**Keywords:** sports education, planification, evaluation, bidirectional feedback, learning, nonlinearity

## Abstract

Quality in education is one of the 17 goals in the United Nations’ sustainable agenda for 2030, presupposing careful planning and assessment of learning. Traditional planning in sports education (either in training or school settings) largely adopts pre-determined learning sequences and temporal milestones that, in theory, enhance the learning process. However, learning is a context-dependent, non-linear process with considerable intra- and interindividual variability, whereby planning and assessment should also be non-linear. In this narrative review, the main findings suggest that the specific teaching or training contents and their relative (i.e., ordering or sequencing) and absolute timing (i.e., the specific time point where certain learning or adaptations are expected) should vary depending on the learners and the context. In a process-oriented perspective, this requires flexible planning and the establishment of ongoing bidirectional links between planning and assessment. In this framework, assessment should be a flexible, evolving, and daily pedagogical tool instead of a set of formal checkpoints. We further explored how planning and assessment could be linked to provide an ongoing feedback loop that respects the individuality of each learner and its context, and therefore hope this review helps bring about a change in current planning and assessment paradigms in sports education.

## 1. Introduction

Quality in education is one of the 17 goals presented in the United Nations’ sustainable agenda 2030. Contradicting the “one-size-fits-all” approach to education, a “tailor-made” or learner-centered approach acknowledges that learners (e.g., students and athletes) and their learning contexts may differ substantially, promoting equity (i.e., individualization) instead of equality (i.e., generalization) [1]. Penney [2] advocates that equity implies giving, celebrating, and considering the social and cultural differences of individuals during educational settings. The achievements of quality in education presuppose a careful planning and assessment of learning, acknowledging the mutable features of each leaner and their dynamic nature [3]. Intra- and interindividual variability (e.g., motivation, sports background, and personal goals) should be considered when planning more adequate teaching/coaching strategies, as well as in the process of assessing learners [4]. In this vein, appropriateness, considered as a pedagogical perspective that intends to help educators to structure and create meaningful learning environments [5,6], is a useful pedagogical concept for designing representative learning environments (i.e., those that hold personal significance [7]) to all sports learners, while respecting intra- and interindividual variability of learning over time [8]. Physical education and competitive sports are contexts of learning and development in which the application of the appropriateness concept can be extremely valuable [3]; however, these ethical and methodological concerns require flexible planning and intertwined assessment processes [9].

Planning is the delineation of the main goals and drafting the potentially appropriated content to address the learner’s individuality at each particular moment [10]. Planning extends over a continuum that ranges from general stipulations (i.e., a declaration of intentions) to more detailed plans (e.g., progression of drills to solve a specific technical issue within a training session) [11,12]. Of note, planning should consider the non-linear and dynamic characteristics of learning [13], embracing intra- and interindividual variability and the largely unpredictable and “messy” reality of teaching and coaching [11,14]. Traditionally, planning stipulates relative strict timelines based on “optimal” or “ideal” sequences for pre-specified contents (i.e., their relative timing or ordering), both in school (i.e., scholar programs/curricula for each schoolyear) [15] and sports contexts (e.g., largely pre-defined training cycles for each season) [16]. This, however, implicitly assumes that learning is a linear and straightforward process. Recently, more flexible approaches to the planning processes have been proposed [13,17,18], arguing for a deep and ongoing dialogue between planning and learning [13,19]. Of course, such dialogue relies heavily on the process of assessment, which is, therefore, deeply intertwined with the concept of planning.

Assessment helps regulate the nature and specificities of the learning process by continually feeding the plans with relevant, real-time information and stimulating adjustments on the teaching or training process [20]. Dare we say that teaching requires training and training does not survive without teaching? Although assessments are usually applied as endpoints (e.g., to grade a student in Physical Education classes), their greatest contribution should be to provide guidance for the learning process [21], which is already common practice in sports training contexts (e.g., the daily monitoring of training loads to adapt the following scheduled training sessions [22]). A major concern is that following standardized assessments (in their contents, form, and/or timing) may not place the learner at the center of the process, emphasizing the misleading idea that “one-size-fits-all” [8]. To exemplify, in a team sport the official match provides a great assessment moment, delivering information about in which areas the players and the team performed well and what must be improved. From this viewpoint, the assessment process provides valuable information (i.e., weaknesses and strengths of learners) for sport practitioners to guide their planning and for learners to understand in which aspect they have to improve (i.e., becoming the practice representative), keeping them motivated to go one step further [23,24]. Still, we acknowledge that punctual and formal moments of assessment could be helpful given their pedagogical power [8]. However, since learners respond to the same stimulus in diverse ways and in distinct timings, we propose that the most relevant assessments should occur on an ongoing basis and not merely at pre-determined moments (e.g., evaluations at the end of each curricular unit) [25,26].

Planning and assessment should therefore be applied in an integrated and flexible fashion, but are often standardized and largely pre-established [12,27]. In this vein, the usefulness of assessment as a regulator of the learning process is largely ignored. At the same time, due to inherent variability of learning, this relationship between planning and assessment aids to regulate learning (e.g., understanding how different players respond to a specific training session and whether the proposed goals and/or means should be re-thought). Assessments should closely follow the evolution of the learning process, using open and flexible tools instead of standardized tests that may not be sensitive to the evolution of the process [20,28]. Bidirectional feedback should be established between planning and assessment to promote equity and raise the quality in sports education. This bidirectional relation would result in planning that is sensitive to the current learning status constantly reflecting on whether it should or should not be adapted considering the timing and features of each learner [13]. In a nutshell, planning and assessment should perhaps be written in pencil, to be easily erased, redesigned, and/or built upon.

This narrative review explored the usefulness of establishing ongoing bidirectional feedbacks between assessment and planning, the interdependency of both within complex, ambiguous, and uncertain educational contexts, under a non-linear perspective that respects the particularities of learners, contexts, and learning processes. Narrative reviews do not aim to provide detailed methodological descriptions and exhaustive description as do systematic reviews [29]. Instead, they provide relevant critical and reflexive perspectives on the literature [30,31], delivering a broad overview on a topic [32,33]. Narrative reviews may lack the systematization of systematic reviews but gain in providing a different organization of the body of evidence and in delivering an insightful conceptual mapping that helps to (re)interpret the existing body of literature [30,32]. Narrative reviews should provide a judicious and careful selection of the literature with a focus on what is relevant to convey the key messages [30], and adopt a structure based on key concepts [32] to better tell a compelling story. Here, the aim was to provide a broad-view conceptualization of a theme [34], contributing to furthering its understanding and organizing its main ideas in novel and/or elucidative ways. We explored how different frameworks in the literature on planning and assessment can be contextualized. At each step of this review, theoretical concerns were sustained with practical examples to reinforce the need to establish bidirectional links between these two dimensions. Firstly, a case was built towards the advantage of adopting flexible perspectives of planning. Secondly, assessment is revisited as a non-linear process, but most importantly as a true pedagogical tool instead of a mere checkpoint. Thirdly, the bidirectional relationships between planning and assessment were explored.

## 2. An Emergent and Flexible Perspective of Planning

Planning processes range from stipulating general goals (i.e., a declaration of intentions) to more detailed outlines (e.g., a pre-defined road map applied to the microcycle or to the training session) [35]. Of note, although a microcycle usually corresponds to a training week, it may encompass time periods shorter or longer than a week [36]. From planning the entire scholar year or sports season to programing the class or training session, the goal is to challenge the learners and stimulate their development [15]. There is a temporal hierarchy from planning (as a macro or long-term viewpoint) to programming (as a micro or short-term viewpoint) [10,13]. Planning focuses on the long-time delineated strategies, general ideas that will guide the learning processes [16]. At the other end of the spectrum, more detailed and short-term applications are detailed through programming [16]. Often, the same learning goal may be achieved using different programming strategies [37,38]. Planning is the over-arching roadmap and may or may not be periodized [11,13], with periodization implying an a priori division of a season into *periods* [16]. It should be clear that lack of periodization does not equate to lack of planning [13]. Against other proposals [39], we contend that planning always precedes periodization and is indeed a more ancient and broader concept [40,41].

It is common for the teacher or coach to *plan* (but not necessarily *periodize*) the school year or sports season in advance, i.e., to pre-stipulate the contents/themes and their sequence for the entire year [42]. On the surface, this may seem an adequate and rational approach that delivers a sense of organization, but ultimately it assumes a pre-determined sequence or ordering of events, contents, and timings that may not fit the reality of the learning process [11,13,43]. Indeed, both the timing and the sequence experienced by each learner could be very distinct [44,45]. The time needed for different learners to acquire certain skills [46], the “optimal” sequence to achieve a specific goal or competence [47], or even the contents and strategies used to teach [48] are likely to vary from learner to learner. Failing to acknowledge this feature of learning largely ignores the within-group variability and assumes the stability and uniformity of learning paths and their rhythms [45], adopting a general plan for all the learners, with only minute adjustments. As an example, Pass et al. [49] recently showed how difficult it was to follow a periodized program in a youth soccer academy through all the complexity inherent in the learning process and need of constant adaptations. The issue is not with planning—which is highly advisable—but with the amount and detail of planning.

In this vein, planning refers to a general declaration of intentions, maintaining a considerable level of flexibility [13], and is probably best devised as a temporary scaffold until better and more up-to-date information is available (which will link to the topic of assessment) [50,51]. Periodization refers to the manipulation of loads and contents in a specific time schedule (calendar) [16,52]. In sports training, “periodization” is a well-known concept largely developed over time, and it tends to be a pre-established way to handle loads and contents in time [16,27]. Matveyev [53] systematically organized and popularized “periodization” as the management of the practice content based on an annual calendar (time), and since then numerous definitions have been used [27]. Many periodization models are conceptualized in literature such as block periodization [54], reverse periodization [55], and daily undulating periodization [56], just to mention a few. All of these approaches are based on a sequence of learning contents and load in an established time frame, assuming that the previous phase will boost the next one (i.e., linearity) [57]. Therefore, a linear framework is operating even in so-called non-linear periodized models whereby certain inputs are expected to produce predictable outputs [13]. According to Kataoka et al. [27], periodization is an organizational approach to training that considers the competing stressors within an athlete’s life and creates “periods” of time dedicated to specific outcomes. However, some works recently questioned some of the assumptions inherent in the periodization concept such as linearity and prediction of performance [27,58,59]. Therefore, periodization may help organizing pedagogical and training processes, but does not necessarily solve the planning-related problems if the mindset is focused on largely predetermined contents, sequences, and timing of events.

Learning should best be approached as a dynamic system [60]. Dynamic systems develop over time according to their initial conditions and the interactions within the variables involved [61]. Essentially, a dynamic system is composed of many interacting elements and exhibits adaptive variability, characterized by a highly complex structure [62,63]. Typically, the behavior of these systems is very difficult to model due to the intertwined and non-linear relationships between its elements [64]. In non-linear systems, minor changes in an input can produce disproportional outputs (i.e., non-proportionality principle), and even in the same organism, the same input can produce different responses over time [65] (e.g., the same training program may produce different outcomes in different players or in the same player in different moments of the season). Humans may be considered dynamic and complex systems [66,67]. In this context, most of the so-called “non-linear plans” still define temporal endpoints [68] or start from the end of the season and go backwards to the beginning [13], i.e., a linear relationship between input and output and a linear path towards the goals is still (even if unconsciously) expected. But since learners and learning environments behave non-linearly [69,70], it should not be assumed that all the learners will respond to the same stimuli at the same time and/or in the same way (i.e., direction and proportion). A “non-linear planning” could thereby be more useful and appropriate in both school and training contexts. Such “non-linear planning” would not predetermine how long each training period should last, nor should it predetermine which contents should be taught next [13].

Indeed, the complex, dynamic, non-linear nature of the learning process underlies the uncertainty of outcome of any sports training and teaching stimulus (e.g., not all the students learn and acquire the same capacities after equal classes) [44]. In this sense, it becomes doubtful that there is an “optimal” or “ideal” order or sequence of events that can be predetermined in sports learning [44]. In such dynamic systems, it is not possible to predict stimuli output in terms of magnitude, timing, or even direction [61]. Additionally, Raw et al. [71] developed the idea of multifactorial interactions (i.e., cognitive and motor) to skill acquisition, not limited to motor abilities. It is important to note that the relevance of generic planning methodologies are challenged by the complex biological systems [58]. For instance, different students participating in the same classes (i.e., same stimuli and context) may not acquire the skills at the same time; different classmates will produce different outcomes from an equal athletic class. Such a non-linear phenomenon in learning and stimuli adaptation makes it hard to predict adaptation and performance for months, weeks, or even days [69,72]. Therefore, the urgency of establishing a continuous, ongoing dialogue between planning and programing processes, as well as between these and assessment, arises due to the need of individual adaptation during the whole learning process [37].

The determination of specific tasks and the manipulation of goals and contents depends on short-time adaptations that may have great impact on long-time planning [37]. For example, the time needed for the students of a physical education class to learn a specific game behavior (for instance, the setter penetration in volleyball) should influence how much deeper in game understanding the class could reach. Programming refers to a micro-level concept of the hierarchy (i.e., the shorter temporal scales), dealing with the micromanagement of training variables such as intensity, volume, and contents within a shorter period of time (e.g., a week of training or even a single class/training session) [10]. For example, in the physical education class, the teacher plans to address passing technique in volleyball and the programing class involves small-sided games to promote higher number of contacts with the ball in that specific situation. However, different learners, such as more analytical ones (i.e., due to their background in the sport), can be more receptive to different approaches and so, different programing strategies could be applied into the same planning. Returning to volleyball in scholar contexts, different players might achieve different abilities in the same class, which can influence further classes and even split the class into different groups [73]. Once again, the flexibility and moldable features must integrate the learning process to better adapt to different learners and/or moments [17].

Importantly, planning and programing are deeply connected and reciprocally influence each other (based on bidirectional feedback between planning and assessment) [74,75]. For instance, the scholar planning could require adjustments according to the responses of the students in the proposed tasks; at the same time, those tasks should be chosen based on the declaration of intentions within the planning [25]. Let us suppose that a class is assessed in the specific content taught or practiced in the previous week (e.g., attacking in volleyball), and the assessment suggests that low efficiency and/or efficacy was achieved; the subsequent week could handle the same content, but maybe using a different approach, or re-think the goals and perhaps place that content in stand-by, focus on different contents, and wait for future opportunities to improve upon the initially planned content. Therefore, constant, ongoing adjustments to the initially idealized planning are useful, adapting to the learners’ development without adhering to strict and previously established timelines [13].

From this perspective, planning in both sports and educational environments should be flexible and adaptable according to different individual characteristics at each moment due to their intrinsic non-linear features [18,76]. However, the flexibility of the plans does not equate to delivering arbitrary, random teaching or training processes, and should be based on a critical analysis of data emerging from the actual learning process [13]. That requires *assessment*. Assessment can act as a bridge between learning moments (i.e., the process as it unfolds) and subsequent planning, being able to guide learning instead of turning the flexibility of planning into a random process. Regular monitoring of training will always deliver a level of forecasting superior to mere guessing [77]. The same should be applied to scholar contexts and competition in sports training. Hence, assessment strategies should build bridges between current planning designs and real performance and learning of athletes and students.

## 3. Assessment: From a Formality to a Relevant Educational Tool

Assessment is a process that supports learning through measurement of the learners’ achievements [78]. It plays a major role in the learning experience, not only to diagnose the major learning deficits (i.e., specific motor skillor sports behavior), but also to regulate and measure the learners’ evolution throughout the process [79,80]. Additionally, assessment might motivate learners providing relevant feedback to learning experience [21]. Coaches and teachers should choose wisely the assessment types and moments as regulators of their practice to better drive the planning, improve it, and customize it to the learner. Different types of assessment have been documented in the literature [21], including *formative* (focused on the assessing the process), *alternative (*the goal is beyond grading the learner in tests or exams and could include activities such as journals, interviews, or portfolios that promote the reflection about the process), *integrated* (assessment as a part of learning process), *learner-centered* (clearly focused on the learners and their evolution), and *summative* (focused on grading the student or athlete) [21]. Formal and punctual assessments are used in specific time points [21] and may be used as a complementary tool to ongoing assessments [81] which should be at the core of assessment processes [7,20,25,26].

To measure the evolution and detect difficulties regarding the student, teachers should not “take just one picture”, but compare different pictures over time [82]. Only ongoing assessments can provide important up-to-date feedback about the players’ readiness for next sessions and hint at their evolution over time [83]. Ongoing assessments are critical in the design of appropriate learning scenarios (i.e., training sessions or classes) [84,85]. Thus, ongoing assessments (regular, instead of punctual assessment moments) are important strategies to detect fragilities in the teaching–learning process and help to define goals to next training session or class, promoting constant adaptations in learning process (i.e., regulator of learning) [86]. Those formal moments tend to be used exclusively to grade students (i.e., summative assessment) at the end of each learning module instead of being used frequently to adjust the learning process (i.e., formative assessment) [21].

Different types of assessments establish a complementary relationship among each other [87] and should dialogue with the planning process to achieve a more detailed and individualized learning process [25], since those different types of assessments provide different insights about the learner building a more robust profile [88]. For instance, a learner-centered process should include formative assessment moments; furthermore, the process becomes formative only when the outcome is used to adapt teaching procedures and strategies [89,90]. Learning contexts could influence not only the type of assessment but also the appropriate moment to be applied [86]. Simultaneously, other kinds of assessment (such as peers-assessment and self-assessment) might complement the collection of information about the learning process to adjust it [26]. This does not exclude summative assessment from the learning process; instead, it generates a diverse assessment environment not exclusively dependent on summative and punctual assessment [20,87]. Those multiple and complementary ways of assessment do not negate the use of standardized tests (in their contents, form, and timing), but restrict their benefits and pedagogical usefulness. In the same vein, the focus of comparing each learner to normative values is perhaps not the most important role for an assessment [89]. For instance, typically learners are assessed in different moments over the scholar year and their outcomes are attributed, measured, and interpreted in comparison with the classmates or the overall benchmarks that are pre-defined [87]. Here, we advocate that a self-comparison of different aspects over time could be more useful and help learners assess their own evolution [91,92].

Assessment for learning could also benefit from moving away from a test-based culture (i.e., overly focusing on grading) to a learning-based culture (i.e., using assessment to enhance learning power) [21,90]. Although this is largely encouraged in the literature [25,28,93], for many years it has not been used in practice, as suggested by a review of US assessment practices [28,90]. To exemplify, traditional assessment tools include highly standardized test batteries such as the EUROFIT^®^, FITNESSGRAM^®^, or UNIFIT ^®^ [94,95]. EUROFIT^®^ recently tried to describe normative values to children aged 7–19 years based on an established protocol [94]. Except for the importance of the determination of normative values, applying the inherent intra- and inter variability to learners from different contexts, the outcome from fitness test score are decontextualized [28]. Established protocols do not usually consider the context of application (i.e., school or sports training; experienced athletes or not) and the content used is pre-established and not adjusted to the specificity of the past learning moments [96]. Teachers should consider the students’ skill level in order to enhance the use of small-sided games [97], and a more flexible approach to assessment tools [98] in sports education is advised.

The assessment tools should be flexible in three main domains: timing, context, and content [99]. In this way, strategies can be designed to develop and assess learning instead of focusing on grading the learner according to pre-established criteria and uniform design (*one-size-fits-all*) [79]. For example, moldable assessment tools as proposed by Atkinson and Brunsdon [98] in volleyball, basketball, and soccer in scholar contexts are adaptable to different moments and evolving learners’ capacities. The application of the same protocol (i.e., rigid assessment tool) will assess the same capacities or skills which ignores the moment of assessment and the evolution to more advanced skills [98]. In this way, even if the assessment is used over time (i.e., ongoing) it allows the teacher to understand learners’ improvements in a specific task and determined context [81]. To date, scientific sports literature presents many useful tools to access learning and performance process in specific sports, to exemplify: Tactical Assessment Instrument in football (TAIS) [100]; Basketball Learning and Performance Assessment Instrument (BALPAI) [101]. Other sports assessment instruments are more general to different sports such as Team Sports Assessment Procedure (TSAP) [102] and Game Performance Assessment Instrument (GPAI) [103]. GPAI is a multidimensional system used in different sports such as volleyball, softball, soccer, or basketball. With this assessment mechanism we are able to code and observe performance behaviors that demonstrate the ability to solve tactical problems in games by making decisions, moving appropriately, and executing skills [103,104]. These tools are flexible in their content (i.e., they can be adapted over time) and can be applied to real game contexts (or to an analytical situation) [103,105].

Furthermore, the assessment tools should also be assessed [106,107]. This idea of “meta-assessment” (i.e., the assessment of the assessment tools and consideration about their utility in planning) is crucial in terms of keeping the assessment open and flexible to different paths [98]. Thus, learning is deeply dependent on planning and assessment and these processes can cooperate with each other to enhance the learner development [74,75]. As stated by Neupert et al. [108] the misconnection between the assessment (even if it is ongoing) and its subsequent application could result in a problem or in insufficient pedagogical connection (i.e., the application of assessment data into planning concerns, such as major difficulties or most recent improvements). If, for example, a proposed task is too easy, the inadaptation of the task and the assessment will not provide any useful insight about the learning process. The same occurs when the task is too difficult or when the criteria are too hard, whereby the students are not able to perform the proposed task and consequently there is no useful information to the subsequent learning moments.

We recognize that sports training and school settings are different contexts, with distinct demands. For instance, school contexts present very distinct realities: more heterogeneous groups, limited contact periods (i.e., established start and ending date and reduced Physical Education hours), and less time in contact between students and teacher [79,107]. However, there are many connections between them (e.g., the importance of assessment in learning regulation). In sports training, ongoing assessment is most commonly used to assess physical demands [109] or tactical behaviors [110]. However, there seems to exist a gap between athletes’ and coaches’ point of view in how those assessments are effectively used in the control of training [108]. In the sports context, the formal competition presents itself as an opportunity to promote ongoing assessment. Here, competition can occur weekly/bi-weekly/monthly, or in some cases, spaced very far apart in time. This characteristic of the sport will influence how ongoing assessment is used. In the case of weekly competition, the assessment is inherent in that moment; in sporadic competition, coaches need to create representative learning scenarios to assess the group more often [84,111].

Those characteristics are inherent in scholar contexts which might promote a few punctual and formal moments of assessment [28]. Attending to the diversity and heterogeny within classes (i.e., big variability within learning groups), the assessment tools should be even more flexible and adapted to the individuality of the learner [79]. For example, utilization of heterogeneous teams that compete in a physical education class, where after each match, all the players reflect about their performance and help each other, enhancing the reflection about the process, the searching for different solutions together, and creating an engagement environment in the class. Ongoing assessment is, in many ways, an upgrade to the learning process that implies time, reflection, and planning [86]. It might be important to inform about problems related to the teaching–learning process that need to be addressed in the subsequent pedagogical interventions [8].

Similar to planning, assessment should best be applied in a non-linear fashion, accepting the variability inherent to the learner and context [12], and in line with a non-linear approach to planning. Understanding assessment as a non-linear and complex process implies that the schedule, tools, aims, and protocols used should be flexible and adaptable over time and according to the group where will be used [26]. In sports training, different season periods might have different demands and that should influence the representativity of the assessment over the time [112,113]. For example, a play-off period of a season when the team always plays against the same opponent is different from the regular phase where the coach must always analyze against diverse opponents; in the latter context, the constraints change more often. In school, the teacher could plan a specific task to the class and consecutive assessment; however, the students correspond significantly well to the proposed task, and this become too easy for them and must be changed. In this case, the assessment tool should be flexible enough to be changed and adapted to the different goal proposed. Hopefully, the importance of ongoing assessment becomes clear in the context of ongoing monitoring of the learners’ performance and deficits in specific timings and contexts. Thus, it is urgent to understand how planning and assessment should dialogue to create a “sweet-spot” to promote a better and valuable cooperation.

## 4. Ongoing Bidirectional Feedback between Assessment and Planning

Assessment and planning should operate in interdependence, as a dynamic and bidirectional system responding to the non-linearity of learning process [8,20]. Assessment identifies specific problems or opportunities (i.e., each assessment is a diagnosis) and helps to regulate the learning process (e.g., setting next goals) [89] by informing the plans. Such interplay of information will provide important inputs to adapt learning scenarios, becoming appropriate for reaching out to every learner [3]. To implement those ideas, both planning and assessment should be approached in a flexible and adaptable fashion, being capable of changing to satisfy the learners’ needs at different time points [25]. As planning and assessment are deeply intertwined within the complexity and unpredictability of the learning process, they should incorporate non-linear features (e.g., withdraw most absolute and relative timelines/ordering). This idea implies a rupture with current applications of assessment is school [42] where the learners are compared with normative values acquired through general assessment tools [94]. In the sports training context, on one hand, the performance of the players and/or team is also frequently compared with pre-established values of reference, or mean values derived from match reports, with the analysis of how that player evolved over the season [114]; on the other hand, the growth of technology applied in sports allow coaches to control players performance and indexes much more individually [115,116].

As stated by Clark [90] and Hay [28], the assessment process as it is commonly applied, guides teachers and coaches into long, complex, and frequently decontextualized outcomes. This is particularly evident in school settings, where the timing, nature, and context of assessment is defined a priori, meaning that the assessment day could fit and be beneficial for one set of learners, but unsuitable and harmful for other learners [117]. On the other hand, the competitive nature of the sport context replaces the vision from assessment to monitoring—which acts as an assessment that is more detailed, frequent, and deeply contextualized. The implementation of flexible and adaptable assessment tools gets a higher importance in this regard than assessing, on an ongoing basis, what it is intended to. Specifically, sport practitioners could intend to assess team or class (macro-level), small groups, such as athletes and non-athletes in a class (meso-level), and individual analysis (micro-level). For instance, scholar contexts have been focused on assessing students to grade them at the end of the semester instead of using different, more useful, and motivating approaches such as (i) modifying constraints and contents of assessment individually and (ii) using different forms of assessment (e.g., same content into different criteria) [25]. In both cases, the assessment should drive the adaptation of the planning process. In contrast with summative assessments at the end of each module/semester, the utilization of those approaches in ongoing assessment allows the teacher/coach to redesign subsequent teaching moments [118].

Concomitantly, sports practitioners and physical education students seek assessment of teaching–learning contents in different ways during the year/season [98,118]. In team sports, the recent growth of technology and its utilization in sports monitoring allows the technical staff to have a much clearer view about players’ performance during a match or a training session [77,119]. This opens the possibility of increasing the representativity of the practice and the most precise direction of each session, such as the comparison of different players’ performance between specific training tasks and the official match [120]. On the other hand, for many years, the assessment of performance in team sports was based on general and decontextualized tests, focusing their attention on individual actions and ignoring collective behavior and connections [102]. The necessity of coaches to receive useful information to enhance pedagogically the learning process forces the investigation to close the gap between the assessment and the real demands of competition contexts.

These features are shared between sports training and school settings, so these assessment characteristics might be adapted in scholar contexts, for instance, using observations that permit teachers not only to define whether the student is doing right or wrong but also to determine the main issues in learning a specific technique or understanding a team behavior [118]. Open, flexible perspectives allow teachers and coaches to change the contents over time. This continuous process of assessment and the ongoing bidirectional links between planning and assessment may truly incorporate the principles of non-linear teaching or coaching. With regard to more punctual, formal assessments, it is possible to start with a standard version of an established assessment tool (e.g., the GPAI) and change it depending on the learners’ evolution and on the context. Our group is currently implementing empirical research addressing these issues where flexible planning and an ongoing assessment, using a flexible tool which is adapted once per month, aims to dialogue each other to enhance learning power—see Figure 1.

## 5. Practical Implications

We suggest that sports coaches program a training week only *after* the previous weekend match (or assessing moment). That planning should be done after a careful interpretation of the performance and learning indexes retrieved through assessment. In the case of non-existing weekly competition, we encourage the creation of an assessment moment to check the principal concepts that worked before and promote moments to assess what has been learned (which already implies a tailor-made instrument, as this is likely to change in time). For example, a volleyball coach teaching young players to jump set may assess the percentage of jump sets were taken within the opportunities that allowed a jump set attempt; if there were 50 opportunities to jump set and the team took 45, it seems that the training content was assimilated and perhaps the coach can plan to move on to another goal or content. However, if the team only took 15 of 50 chances to apply the jump set, perhaps it is advisable to keep developing this content. At a more micro scale, every training session should be flexible enough to be adjusted after the previous one (i.e., what happens in one training session may have consequences for what should be performed in the next). For instance, if the coach notes difficulties in proposed tasks and the established goals were not achieved, the next session should be planned to taking into account those issues. The events of one training session may even suggest that certain goals should be re-thought.

In school settings, we recommend the ongoing application and adaptation of tools instead of fixed use of standardized tests at predetermined moments. The class is generally formed by heterogeneous students with distinct genetics, backgrounds, and experiences. These contextual features will influence not only the starting point (importance of diagnostic assessments), but also the rhythm, direction, and magnitude of learning (ongoing assessment) [45]. It is therefore primordial to adjust the planning according to the characteristics of the group (i.e., past experience of students in sports: are they athletes or not? In which level?) and subsequently assessment moments and contexts. To exemplify, we suggest that the teachers: (i) stimulate continuous self-assessment during and after the classes as a reflective moment of learning; (ii) promote peer-assessment to increase accountability and awareness; and (iii) change the format of the assessment as well as its contents and constraints as the process unfolds and evolves. For instance, it might be useful to create different groups within a Physical Education class, each with a mixed level of competence, attributing different roles within the group (e.g., a more experience student acting as a coach/teacher) to stimulate reflection, engagement, and deliver greater variability in feedback type, content, and style.

In both contexts, the implementation of different forms and types of ongoing assessment is crucial if we search for valid insights to regulate the learning process. We encourage coaches and teachers to: (i) evaluate the starting point of teams and classes with diagnostic assessments; (ii) define general goals for the season/year based on that diagnostic, but avoid entering into excessive detail and be open to change; (iii) program in detail only for short periods of time ahead (perhaps two weeks at a maximum, but often less than that); (iv) regularly assess learning in different ways (self-assessment and peer-assessment); (v) use the outcome of regular assessments to change the over-arching plans and/or the more immediate programs (if necessary, readjust even the major goals); and (vi) use different assessment strategies according to the moment of the year/season. This ongoing bidirectional feedback between planning and assessment will provide useful insights to help guide what should be a learner-centered process.

## 6. Limitations

There is still a gap in research about bidirectional feedback between assessment and planning. To deeply understand the link between assessment and planning, the theoretical proposals we have devised here need to be tested in the field. Both in school and sports contexts, an implementation of flexible plans, coupled with ongoing, flexible, and adaptative assessment tools should be delivered. At a minimum, it is important to assess the feasibility of these proposals, testing its application in different contexts and identifying the opportunities and barriers that emerge, as well as the perceptions of the relevant stakeholders (e.g., students/athletes, teachers/coaches, managers, parents). Ideally, these interventions consist of flexible and coupled planning and assessment processes should be compared with more traditional planning models and to the more common standardized tests.

## 7. Conclusions

Sports education commonly refers to the relevance of planning and assessment, but here we highlighted the importance of intertwining planning and assessment to enrich the pedagogical process. Three main points were put forth: (i) the need to establish flexible approaches to planning; (ii) the need to establish flexible assessment processes and tools; and (iii) the need to establish ongoing bidirectional feedback between planning and assessment. Our review strongly suggests that planning and assessment can be adjusted over time to respond to the intra- and interindividual features and needs of each learner and to respond appropriately to a dynamic, complex, and non-linear learning process. If we pursue quality in sports education, it is perhaps time to look at each learner individually and think about how to potentiate their learning path (which requests a daily assessment/monitoring and (re)planning), instead of first looking globally to a group, measuring means, and inferring how close are we from standardized. Despite the theoretical contributions provided in this review, from a practical perspective the integration of planning and assessment is still scarcely explored scientifically, especially as assessments tend to be standardized and stable in time and the plans are largely predetermined. For this reason, its impacts on learning development remains ambiguous and unclear, but our research group is already engaging in empirical research focused on this research question.

## Figures and Tables

**Figure 1 ijerph-19-12068-f001:**
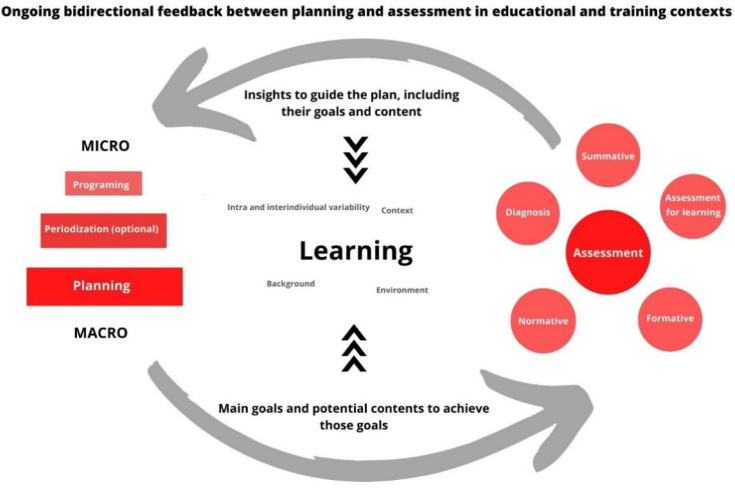
Ongoing feedback between assessment and planning.

## Data Availability

Not applicable.

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
