# Peer review of "Ongoing Bidirectional Feedback between Planning and Assessment in Educational Contexts: A Narrative Review"

_ijerph, 2022, doi:10.3390/ijerph191912068_

Round 1

Reviewer 1 Report

The paper reflects on the relationship between planning and assessment in the field of sports education. The authors argue for flexible planning and the establishment of ongoing bidirectional link between planning and assessment. The present relevant and recent literature to argue for this narrative review work.

The paper is well structured and organised. Appropriate literature and definitions of key concepts, are included with apt explanations in the filed of sports education. It is an interesting piece of work that touched on the least researched filed of sports education. As the authors suggest, field work is missing to test the theories and enhance the suggested model of learning. Perhaps more researchers can pick this up and conduct follow-up case studies.

Minor typos and editing on lines: 201, 319.

Author Response

Dear reviewer, thank you for taking the time to review our paper. We thank you for the kind comments and have corrected the minor typos and editing on the indicated lines, signaled through the Track Changes function of MS Word.

Kind regards

Reviewer 2 Report

This paper provides a narrative review of the planning and assessment studies to reveal "how different frameworks in the literature on planning and assessment can be contextualized". The paper is well structured, and the arguments are sound. The following are some aspects for the authors to consider and revise in order to make the manuscript clearer to the readers.

1. The narrative review is a new concept to me, and in the manuscript its difference from the systematic review is briefly illustrated. I think the authors can explain further why a narrative review is crucial for this topic.

2. The title "ongoing bidirectional feedback between planning and assessment in educational contexts" gives me an impression that "bidirectional feedback" is a keyword in the text. However, it is not highlighted in the abstract nor listed as a keyword. Please check whether a keyword needs to be added. Moreover, I notice that the review is mainly concerned with the area of sports education. If it is so, the authors may need to consider whether it is necessary to specify this research area. I'm not sure whether the arguments in the manuscript can be generalized to all educational fields or contexts.

3. The Abstract is not well written, as it gives mostly the background of the current review study. What are the main findings, and what is the significance of such a study? These points also deserve mentioning in the Abstract.

 4. In the conclusion part, it would be better to give a brief summary of the main contents of this study and highlight your contributions. 

5. The English needs to be proofread again.

Hope these points are useful for the authors to reconsider and further improve the overall quality of the article.  

Author Response

Dear reviewer, thank you for taking the time to review our paper. We thank you for the kind comments. In response to your specific comments:

1 – In the last paragraph of the introduction, we have expanded on the justification of using a narrative review. All edits were marked using MS Word Track Changes function.

2 – The expressions “bidirectional feedback” and “sports education” were added to the keywords and abstract. These ideas are now also more explicitly expressed in the conclusions.

3 – Actually, much of the abstract does reflect the development and main conclusions of the review, but we acknowledge that our writing was not very clear and thank the reviewer for noting that. We edited portions of the abstract to increase clarity and also tried to emphasize this work’s significance. We hope this improved version meets the reviewer’s expectations.

4 – We agree with the reviewer and have edited the conclusions to improve the sequence and clarity of the main points.

5 – We tried improving the English across the manuscript.

Kind regards